

# COVID-19 and malaria co-infection: a systematic review of clinical outcomes in endemic areas

Amal H. Mohamed[1], Ebtihal Eltyeb[1], Badria Said[1], Raga Eltayeb[1], Abdullah Algaissi[1], Didier Hober[2] and Abdulaziz H. Alhazmi[1]

[1] Jazan University, Jazan, Saudi Arabia
[2] Univ Lille, CHU Lille Laboratoire de Virologie ULR3610, Lille, France

## ABSTRACT

**Background:** COVID-19 and malaria cause significant morbidity and mortality globally. Co-infection of these diseases can worsen their impact on public health. This review aims to synthesize literature on the clinical outcomes of COVID-19 and malaria co-infection to develop effective prevention and treatment strategies.

**Methods:** A comprehensive literature search was conducted using MeSH terms and keywords from the start of the COVID-19 pandemic to January 2023. The review included original articles on COVID-19 and malaria co-infection, evaluating their methodological quality and certainty of evidence. It was registered in PROSPERO (CRD42023393562).

**Results:** Out of 1,596 screened articles, 19 met the inclusion criteria. These studies involved 2,810 patients, 618 of whom had COVID-19 and malaria co-infection. Plasmodium falciparum and vivax were identified as causative organisms in six studies. Hospital admission ranged from three to 18 days. Nine studies associated co-infection with severe disease, ICU admission, assisted ventilation, and related complications. One study reported 6% ICU admission, and mortality rates of 3%, 9.4%, and 40.4% were observed in four studies. Estimated crude mortality rates were 10.71 and 5.87 per 1,000 person-days for patients with and without concurrent malaria, respectively. Common co-morbidities included Diabetes mellitus, hypertension, cardiovascular diseases, and respiratory disorders.

**Conclusion:** Most patients with COVID-19 and malaria co-infection experienced short-term hospitalization and mild to moderate disease severity. However, at presentation, co-morbidities and severe malaria were significantly associated with higher mortality or worse clinical outcomes. These findings emphasize the importance of early detection, prompt treatment, and close monitoring of patients with COVID-19 and malaria co-infection.

# INTRODUCTION

Malaria persists as a severe public health problem in many developing countries, causing morbidity and mortality in residents of endemic areas, particularly pregnant women and young children (*World Health Organization, 2023*). The World Health Organization

Corresponding author
Abdulaziz H. Alhazmi,
hazmi_6@hotmail.com

(WHO) malaria reports of 2021 state that Africa accounted for the highest number of global malaria deaths, with children, particularly those under 5 years, accounting for almost these deaths (*World Health Organization, 2023*). Most tropical countries have endemic malaria, brought on by the Plasmodium parasite, where socioeconomic and environmental variables contribute to its enduring persistence (*CDC, 2021*). Despite the high prevalence of malaria in these endemic areas, most of the population has some level of immunity due to a tenuous, repeated lifetime interaction with the Plasmodium parasite (*Crompton et al., 2014*).

The emergence of the corona virus disease of 2019 (COVID-19) pandemic indirectly increased the prevalence of malaria as the COVID-19 pandemic resulted in disruptions of health services in sub-Saharan Africa (*Gao et al., 2023*). The quarantine measures disturb the continuity of the global malaria programs, the seasonal malaria chemoprevention, and the insecticide-treated bed nets distribution (*Gao et al., 2023*). Morbidity and mortality continue to be of interest globally, particularly in pandemics like COVID-19, which have overburdened healthcare systems and were therefore anticipated to have severe pandemic effects (*Filip et al., 2022*). COVID-19 and malaria have early symptoms that overlap, such as fever, headache, nausea, and muscular discomfort or weariness. These two diseases also have some shared concepts in the patient's immune responses, such as cytokines and chemokines that are rapidly released during COVID-19 and Plasmodium infections, which are essential to the pathophysiology of the diseases (*Chaturvedi et al., 2022*). The co-infection of these two diseases can pose significant challenges for healthcare systems and patients, as both diseases can cause severe illness and complications. Studies have shown that the cases of co-infection of malaria and COVID-19 increased, and individuals with this co-infection may experience more severe symptoms and have a higher risk of complications. These risks are particularly concerning in regions where both diseases are endemic, as they can strain healthcare resources and lead to poorer patient service outcomes (*Alhaddad et al., 2023*; *Carrión-Nessi et al., 2023*).

Ecological research has demonstrated slightly earlier in the pandemic that COVID-19 infection rates and case fatality ratios tended to be lower in countries where malaria is endemic, and according to this research, COVID-19 treatment and prophylaxis may benefit from anti-malarial medications (such as hydroxychloroquine) (*Woodford et al., 2022*). In March 2020, chloroqiune was added to the protocol of COVID-19 treatment by the National Health Committee (NHC) in China (*Woodford et al., 2022*; *Osei et al., 2022*). Nevertheless, the National Institute of Health (NIH) advises against using chloroquine or hydroxychloroquine with or without azithromycin for the therapy of COVID-19 in hospitalized patients as well as in non-hospitalized patients later on, based on data from numerous clinical trials, observational studies, and single-arm research (*Osei et al., 2022*; *Brown et al., 2020*). Irrespective of the severity of the illness, WHO strongly recommends against using chloroquine or hydroxychloroquine in patients with COVID-19 (*Napoli & Nioi, 2020*).

An evidence-based approach to understanding the clinical outcomes of the COVID-19 pandemic and malaria co-infection was carried out to support efforts to mitigate the

clinical outcomes and allocate resources accordingly. The main objective of this systematic review is to assess the clinical outcomes of COVID-19 and malaria co-infection, namely the mortality rate, admission rate to the intensive care unit (ICU), clinical severity, and length of stay at the hospital.

## MATERIALS AND METHODS

### Study design and registration

This systematic review evaluated the short clinical outcomes of COVID-19 in patients with malaria. In addition, similarities and differences in existing evidence and literature were addressed to reach conclusive results. The International Prospective Registry of Systematic Reviews (PROSPERO) registered this systematic review protocol on 23 January 2023 under registration number CRD42023393562.

### Search strategy

We carried out an electronic literature search, including articles on COVID-19 and Malaria in PubMed, ScienceDirect, Scopus, and EMBASE, from the beginning of COVID-19 in December 2019 to January 2023. A structured format based on Preferred Reporting Items for Systematic Review and Meta-Analyses (PRISMA) Guidelines and a checklist were used to select and review studies included in the review (*Page et al., 2021*). We performed the Medical Subjects Heading (MeSH Database) and keywords search for non-MeSH data. Search terms included COVID-19, SARS-CoV-2, Co-infection, Malaria, Plasmodium, Falciparum, Vivax, outcomes, mortality, ICU admission, and morbidity. A manual search for identified references of included studies, relevant reviews, and grey literature was performed to find further relevant studies not found by database search.

### Study selection and eligibility criteria

The inclusion criteria for this review were original observational research (cohort studies, case reports, and case series) published only in English, within the abovementioned time, conducted on co-infection of COVID-19 and malaria, and including all age groups. We excluded systematic reviews, scoping and narrative reviews, review articles, non-relevant articles, and studies not fulfilling the eligibility criteria. COVID-19 and malaria con-infection were defined by a confirmed positive severe acute respiratory syndrome corona virus 2 (SARS-CoV-2) viral reverse transcription-polymerase chain reaction (RT-PCR), or radiological findings compatible with COVID-19, after exclusion of another differential diagnosis, which had at least one episode of malaria simultaneously or after the infection by COVID-19.

Malaria infection is diagnosed based on clinical presentation and parasitological diagnosis with either light microscopy (thin and thick film) or immune-chromatographic rapid diagnostic tests (RDTs) (*CDC, 2021*). The primary outcomes were mortality rate, ICU admission, clinical severity, and duration of hospitalization of COVID-19 patients with malaria co-infection. The other outcomes, like the prevalence of the coexistence of COVID-19 with malaria and related risk factors, were also assessed.

## Data extraction and management

Two authors (A.H.M, E.E.E.) screened the titles and abstracts, and then all authors discussed the full text for the inclusion criteria and any disagreement for study selections. Studies that do not meet the eligibility criteria are documented with reasons and then excluded. Data was extracted manually and transferred to data extraction form, including the following: First name of the author, year of publication, location, study design, the total number of participants, and sample size, including the number of patients who were diagnosed with both Malaria and COVID19, Characteristics of the participants (age, gender, associated co-morbidities), methods of diagnosis, type of plasmodium, and description of the measured outcomes.

## Synthesis of the evidence

Joanna Briggs Institute's (JBI) critical appraisal checklist for observational studies is used to evaluate the quality of the studies, reliability, validity, and relevance to practice (*JBI, 2020*). For certainty of the evidence, the Grading of Recommendations Assessment, Development and Evaluation (GRADE) working group grades for evidence used and rated as high, moderate, low, and very low certainty. Subgroup and stratified analyses were considered according to age groups, genders, and Plasmodium species. In addition, the risks of publishing bias were checked. The Synthesis Without Meta-analysis (SWiM) guideline ensures transparent reporting for reviews as alternative synthesis methodologies to investigate, describe, and interpret critical data regarding the clinical outcomes of COVID-19 and malaria co-infections. A narrative synthesis was performed under the SWIM in systematic review reporting criteria (*Campbell et al., 2020*).

# RESULTS

## Search result

A total number of 1,596 articles were identified through a systematic search based on PRISMA guidelines from four databases: PubMed ($n = 242$), Scopus ($n = 117$), EMBASE ($n = 1,122$), and ScienceDirect ($n = 115$). All identified papers were managed manually, and 1,405 articles were excluded for duplication ($n = 1,064$) or ineligible by automated tools ($n = 341$); 186 of them were not original articles; no associated data in 72 papers, 80 articles were not full article, while three research not include quantitative analysis. Screening for titles and abstracts was conducted for ($n = 191$), and accordingly, ($n = 156$) were excluded. Thirty-five full-text papers were extracted for a more comprehensive evaluation, 19 articles were included in the systematic review, and 16 were excluded as they did not fulfill the inclusion criteria. Table 1 showed the significant excluded studies with reasons for exclusion: four studies due to non-compatible objectives. *Campbell et al. (2020)*, *Bylicka-Szczepanowska & Korzeniewski (2022)*, *Zhu et al. (2022)*, *Arshad et al. (2020)*, seven studies with non-compatible study design (*Zhu et al., 2022*; *Zhang et al., 2022*; *Simon-Oke, Awosolu & Odeyemi, 2023*; *Igala et al., 2022*; *Kalungi et al., 2021*; *EBioMedicine, 2021*; *Zawawi et al., 2020*), other four with non-compatible outcomes (*Arshad et al., 2020*; *Dorkenoo et al., 2022*; *Briggs et al., 2022*; *Wilairatana et al., 2021*), and single study the publication language is not English (*López-Farfán et al., 2022*).

**Table 1 COVID-19 and malaria excluded studies.**

| Author | Study design | Country | Reason of exclusion | The aim of the study |
|---|---|---|---|---|
| *Bylicka-Szczepanowska & Korzeniewski (2022)* | Cohort study | Central African Republic | Different objectives | Assess the prevalence of asymptomatic malaria cases in children and adults living in the DzangaSangha region in the Central African Republic (CAR) during the COVID-19 pandemic |
| *Zhu et al. (2022)* | Cohort study | Shanghai/China | Different objectives | To understand the epidemiological characteristics of imported malaria in Shanghai specifically during the epidemic period of novel corona-virus pneumonia (COVID-19), |
| *Mbah et al. (2022)* | Comparative cross-sectional community study Comparative Study (ecology) | Cameroon | Different objectives | Comparative study of asymptomatic malaria in a forest rural and depleted forest urban setting during a low malaria transmission and COVID-19 pandemic period |
| *Arshad et al. (2020)* | Review article | – | Different study design/objective | The potential role of CD-147, and potential malaria-induced immunity and polymorphisms in COVID-19 patients. |
| *Zhang et al. (2022)* | Cohort study | China | Different objective | Surveillance and response to imported malaria during the COVID-19 Epidemic |
| *Dorkenoo et al. (2022)* | A cross-sectional study | Togo | Different outcomes | Estimate the prevalence of malaria and COVID-19 by PCR and serological tests in febrile patients |
| *Briggs et al. (2022)* | Cohort study | Uganda | Study design | Preprint article |
| *López-Farfán et al. (2022)* | Cohort study | Burkina Faso | Different outcome | Estimate the frequency of SARS-CoV-2/malaria co-infection for the same period |
| *Simon-Oke, Awosolu & Odeyemi (2023)* | Cohort study | Nigeria | Different outcome | Determine the prevalence of malaria and COVID-19 in Akure |
| *Igala et al. (2022)* | prospective, observational study | Gabon | Language | The study published in French |
| *Kalungi et al. (2021)* | Review | – | Study design | Mini review article |
| *EBioMedicine (2021)* | Review | – | Study design | Review of the literature |
| *Zawawi et al. (2020)* | Review | – | Study design | Review of the literature |
| *Wilairatana et al. (2021)* | Systematic review and metaanalysis | – | Study design and different outcomes | Prevalence and characteristics of malaria among COVID-19 individuals: A systematic review, meta-analysis, and analysis of case reports |
| *Konozy et al. (2022)* | Review | – | Study design | Review of the literature |
| *Guha et al. (2021)* | Pilot/Cohort study | India | Different outcomes | To determine the incidence of SARS-CoV-2 infection among febrile patients attending a malaria clinic |

The identification and screening process sequence was displayed in the PRISMA Flow Diagram (Fig. 1).

## Characteristics of included studies

Nineteen studies, with a total of 2,810 participants, 618 of which had COVID-19 and malaria co-infection published from December 2019 to January 2023, relevant to the review question and fulfilling the inclusion criteria, were included in this review (Table 2). The study was based on findings from throughout the world (Fig. 2), including data from

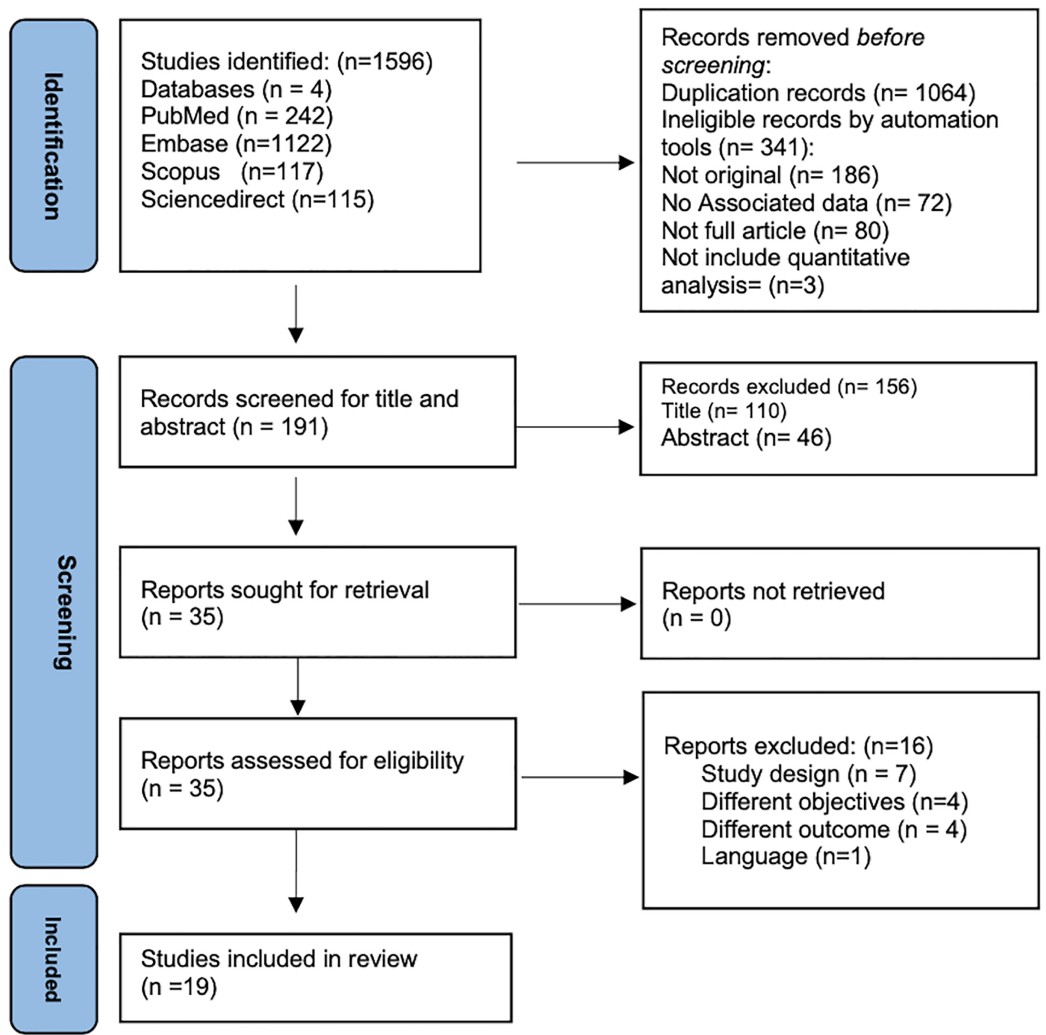

**Figure 1 PRISMA flow diagram for search strategy process.**

Asia (*n* = 11), nine of them from Southeast Asia and one from Turkey, Africa (*n* = 4), Europe (*n* = 2), and South America (*n* = 2). RT-PCR was used to diagnose COVID-19 in all papers, and RDT and blood smears were the preferred investigations for Malaria diagnosis. Three studies used RT-PCR in addition to blood smear (*Konozy et al., 2022*; *Muyinda et al., 2022*; *Caglar et al., 2021*). *Plasmodium falciparum* and *vivax* were identified as causative types in six studies for each, respectively (*Konozy et al., 2022*; *Guha et al., 2021*; *Huang et al., 2022*; *Boonyarangka et al., 2022*; *Forero-Peña et al., 2022*; *Achan et al., 2022*; *Hussein et al., 2022*; *Scalisi et al., 2022*; *Ahmed et al., 2022*; *Caglar et al., 2021*; *Indari et al., 2021*; *Chen et al., 2021*), while in three studies, both species were detected (*Lee, Hong & Kim, 2022*; *Muyinda et al., 2022*; *Suarez et al., 2022*). *Plasmodium ovale* was reported in two other studies (*Shahid et al., 2021*; *Pusparani, Henrina & Cahyadi, 2021*), and one study

**Table 2 Characteristics of included studies.**

| Author | Study design | Country | Sample size | Characteristics of the participants | | | | Malaria Spp. |
|---|---|---|---|---|---|---|---|---|
| | | | | Age | No. of patients with COVID 19 & Malaria co-infection | Diagnosis of COVID19 | Diagnosis of malaria | |
| Huang et al. (2022) | Case report | China | 1 | Not mentioned | 1 | RT-PCR | PCR & Blood smear | P. falciparum (Re-infection) |
| Boonyarangka et al. (2022) | Case report | Thailand | 1 | 25 | 1 | RT-PCR | RDT & Blood smear | P. vivax |
| Forero-Peña et al. (2022) | Case series | Venezuela | 12 (one a pregnant woman and one was an infant) | Mean age 42 ± 18 | 12 | RT-PCR | Blood smear | P. vivax 3 relapse & 6 re-infection cases |
| Achan et al. (2022) | Exploratory prospective, cohort study | Uganda | 597 | Median age was 36 | 70 | PCR | RDTs & Blood smears. | P. falciparum |
| Lee, Hong & Kim (2022) | Case report | Korea | 1 | 63 | 1 | PCR | Blood smear & PCR | P. falciparum |
| Asmarawati et al. (2022) | Case report | Indonesia | 1 | 23 | 1 | RT-PCR | RDT & blood smear | P. vivax (relapse) |
| Hussein et al. (2022) | Retrospective cohort study | Sudan | 591 | 58 ± 16.2 | 270 | RT-PCR | RDT& blood smear | P. falciparum and P. vivax. |
| Muyinda et al. (2022) | Retrospective cohort study | Uganda | 968 | Median age of the participants was 52 | 70 | PCR or RDT | – | – |
| Suarez et al. (2022) | Case report | Venezuela | 1 | 69 | 1 | PCR | Blood smear | P. Falciparum |
| Scalisi et al. (2022) | Case report | Italy | 1 | 8 | 1 | PCR | Blood smear & PCR | P. falciparum & P. vivax |
| Ahmed et al. (2022) | Retrospective cross-sectional study | Sudan | 156 | 65.2 ± 14.5 | 156 | RT-PCR | Blood smear | P. Falciparum only one with P. vivax |
| Shahid et al. (2021) | Case report | India | 1 | 55 | 1 | RT-PCR | Blood smear | P. vivax |
| Jochum et al. (2021) | Case reports | Germany | 1 | 61 | 1 | RT-PCR | Blood smear | P. falciparum |
| Caglar et al. (2021) | Case report | Turkey | 1 | 38 | 1 | PCR | Blood smear | P. ovale |
| Pusparani, Henrina & Cahyadi (2021) | Case report | Indonesia | 1 | 24 | 1 | PCR | RDT | Non-falciparum malaria (recurrent malaria) |
| Indari et al. (2021) | Case report | India | 1 | 28 | 1 | RT-PCR | RDT | P. falciparum |
| Chen et al. (2021) | Case report | China | 1 | 47 | 1 | RT-PCR | RDT & blood smear | P. ovale (relapse) |
| Kishore et al. (2020) | Case report | India | 1 | 10 years old | 1 | RT-PCR | RDT & blood smear | P. vivax (relapse) |
| Mahajan et al. (2020) | Retrospective study | India | 491 | Median age 32 | 27 | RT-PCR | RDT & blood smear | P. vivax |

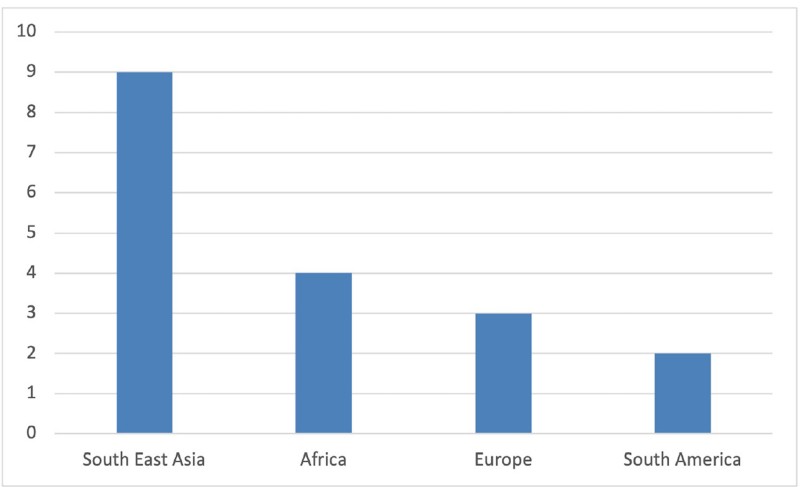

**Figure 2 Number of studies by its geographical distribution.**

reported it as non-*falciparum spp*. Four studies reported malaria infection as relapse, recurrent reinfection, or reactivation (*Konozy et al., 2022*; *Huang et al., 2022*; *Jochum et al., 2021*; *Indari et al., 2021*). As there is no age restriction, all age groups were included. *Scalisi et al. (2022)*, *Muyinda et al. (2022)*, *Kishore et al. (2020)* and *Indari et al. (2021)* reported the outcomes of COVID-19 and malaria infection in pediatric patients; however, in one study, age was not specified (*Konozy et al., 2022*). In the case series, one patient was an infant, and one was pregnant (*Huang et al., 2022*).

## COVID-19 and malaria co-infection; co-morbidities and clinical outcomes

Regarding the outcomes, the duration of hospital stay ranged from 3 to 18 days. However, the disease ran a benign course in seven studies (*Konozy et al., 2022*; *Guha et al., 2021*; *Forero-Peña et al., 2022*; *Hussein et al., 2022*; *Scalisi et al., 2022*; *Pusparani, Henrina & Cahyadi, 2021*; *Indari et al., 2021*), and patients were discharged with a complete recovery. Furthermore, in their cohort study in Uganda observed that patients with COVID-19 and Malaria co-infection had faster recovery and a lower admission rate than those without malaria (*Asmarawati et al., 2022*).

Co-infection associated with severe disease, ICU admission and assisted ventilation, and other related complications such as cerebral malaria, neuro-cognitive disturbance, coagulopathy, and liver derangement was reported in nine studies (*Huang et al., 2022*; *Boonyarangka et al., 2022*; *Achan et al., 2022*; *Lee, Hong & Kim, 2022*; *Suarez et al., 2022*; *Ahmed et al., 2022*; *Shahid et al., 2021*; *Jochum et al., 2021*; *Caglar et al., 2021*). Multisystem inflammatory syndrome (MIS) was reported in an eight-year-old girl in Italy and showed overall improvement in her condition (*Muyinda et al., 2022*).

One study reported 6% of ICU admissions (*Boonyarangka et al., 2022*), while death was seen in three studies, with mortality rates of 3%, 9.4%, and 40.4%, respectively

**Table 3 Comorbidites, duration of hospitalization and outcomes.**

| Author | Study design | Sample size | Co-morbidities | Duration of hospitalization/ days | Outcomes | Comment |
|---|---|---|---|---|---|---|
| *Huang et al. (2022)* | Case report | 1 | None | 18 | Discharged with stable vital signs and mental state | |
| *Boonyarangka et al. (2022)* | Case report | 1/1 | None | 3 | Discharge home with complete recovery | |
| *Forero-Peña et al. (2022)* | Case series | 12/12 | Not mentioned | Mean 11.5 ± 7.2 | 9/12 had moderate to severe COVID-19 disease. | One was a pregnant woman, and one was an infant both were discharge well |
| | | | | | 3/12 had mild COVID-19 | |
| | | | | | 7/12 had elevated AST/ALT levels. | |
| | | | | | 5/12 had thrombocytopenia. | |
| | | | | | 2/12 had elevated creatinine levels | |
| | | | | | 1/12 had severe anemia. | |
| | | | | | 1/12 had thrombocytosis. | |
| *Achan et al. (2022)* | Exploratory prospective, cohort study | 70/597 | 16% Hypertension | Mean 17·4 ± 4·6 | Neurocognitive disturbance | |
| | | | 10% DM | | 86% Discharge in good condition | |
| | | | 7% HIV | | 6% Admitted to ICU | |
| | | | 4% Obesity | | 3% Died 3% | |
| | | | 3% heart disease | | 1% Discharged with sequel. | |
| | | | 1% COPD | | | |
| *Lee, Hong & Kim (2022)* | Case report | 1/1 | Hypertension and dyslipidemia | Not specified | Very benign course | |
| *Asmarawati et al. (2022)* | Case report | 1/1 | Not mentioned | 6 | Prolong fever | |
| | | | | | Hyper-coagulopathy | |
| *Hussein et al. (2022)* | Retrospective cohort study | 270/591 | 41.1% DM | Median 21 | Crude mortality rates: 10.71 per 1,000 person-days | |
| | | | 22.3% Hypertension | | 14% lung edema | |
| | | | 15% CVS | | Renal impairment | |
| | | | 8.5% Neurological disease | | Acidosis, prostration | |
| | | | 8.1% renal disease | | | |
| | | | 3% Malignancy | | | |
| *Muyinda et al. (2022)* | Retrospective cohort study | 70/968 | 28.2% Hypertension 15.9% DM | Not specified | 65% shorter survival lower rates of hospitalization | Co-morbidities for all participants but not specified for Malaria group |
| | | | | | 9.4% dead | |
| *Suarez et al. (2022)* | Case report | 1/1 | Not mentioned | 15 | Discharged | |
| *Scalisi et al. (2022)* | Case report | 1/1 | Risperidone-treated autism | Not specified | Multisystem inflammatory syndrome (MIS), anemia | *Two types of Malaria (P. falciparum & P. vivax) plus COVID19* |
| | | | | | Overall improvement in the patient's clinical conditions. | |

| Author | Study design | Sample size | Co-morbidities | Duration of hospitalization/ days | Outcomes | Comment |
|---|---|---|---|---|---|---|
| Ahmed et al. (2022) | Retrospective cross-sectional study | 156/156 | 38.5% DM<br>37.2% Hypertension<br><br>4.5% Asthma<br>3.2% cancer<br>1.3% COPD<br>1.3% recent surgery<br>0.6% Immunodeficiency<br>Others 29.5% (not specified) | 7.0 ± 5.3 | Overall mortality was 40.4%.<br>Other outcome: Acute respiratory distress syndrome (35.3%) | |
| Shahid et al. (2021) | Case report | 1/1 | DM | 14 | Discharged with complete recovery. | |
| Jochum et al. (2021) | Case reports | 1/1 | None | Not specified | Severe malaria<br>Severe thrombocytopenia Discharged after complete recovery. | |
| Caglar et al. (2021) | Case report | 1/1 | None | 8 | Hyper-inflammatory syndrome<br>Complicated malaria discharged | |
| Pusparani, Henrina & Cahyadi (2021) | Case report | 1/1 | None | 11 | Severe malaria<br>Thrombocytopenia<br>Discharged | |
| Indari et al. (2021) | Case report | 1/1 | None | 4 | Severe hypoxia<br>Cerebral malaria.<br>The patient died | |
| Chen et al. (2021) | Case report | 1/1 | Not mentioned | 13 | Mild disease<br>Discharged with full recovery | |
| Kishore et al. (2020) | Case report | 1/1 | None | 14 | Discharged with complete recovery. | |
| Mahajan et al. (2020) | Retrospective study | 27/491 | 11% Hypertension<br>8% DM diabetes<br>4% Asthma<br>2.8% hypothyroidism<br>1.3% IHD<br>1.1% tuberculosis<br>1.7% Other comorbidity<br>6.5% more than 1 comorbidity | Mean 7.7 | The recovery of SARS-CoV-2 infection in HCWs was faster (mean 8 days) with co-infection of malaria than without malaria | |

 

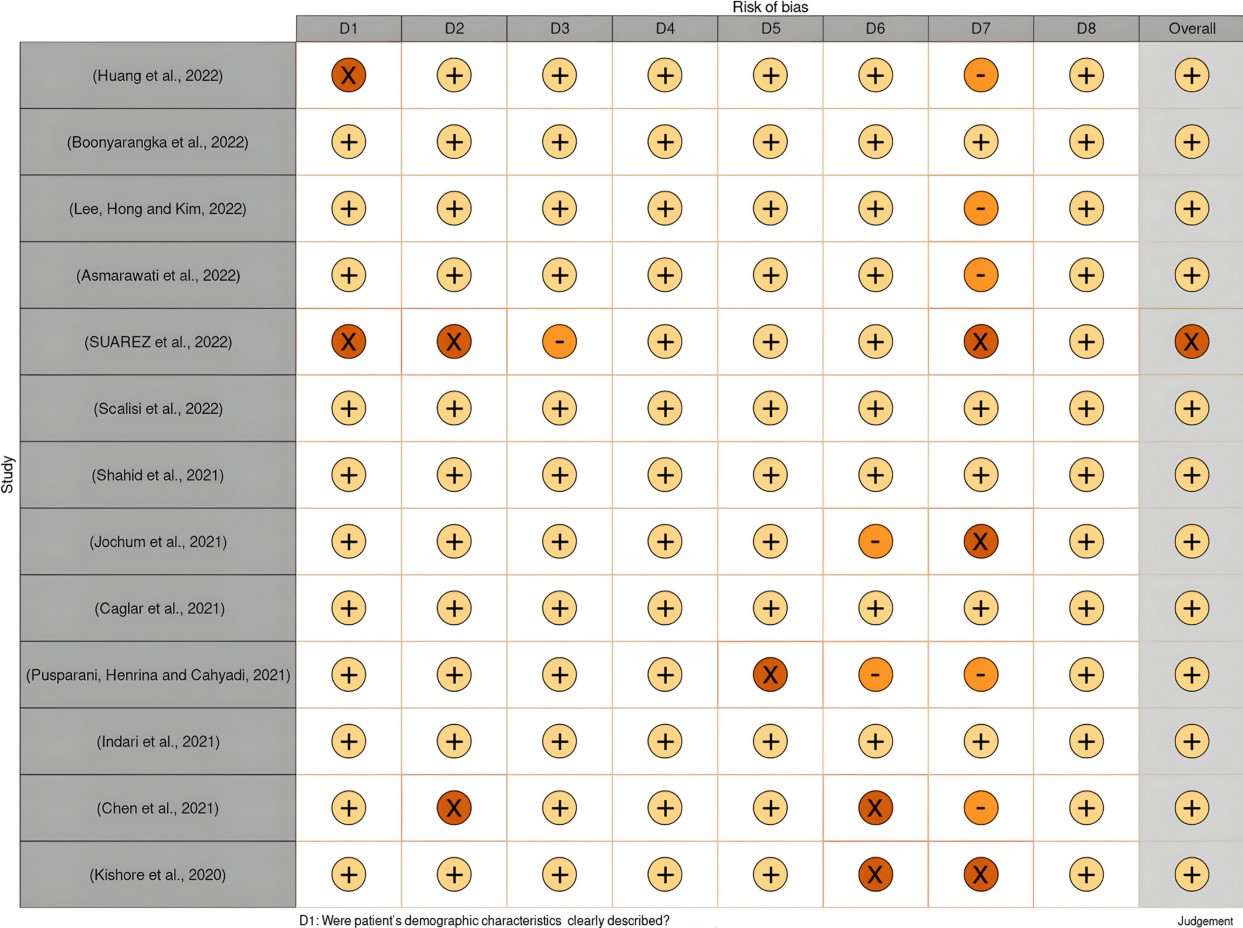

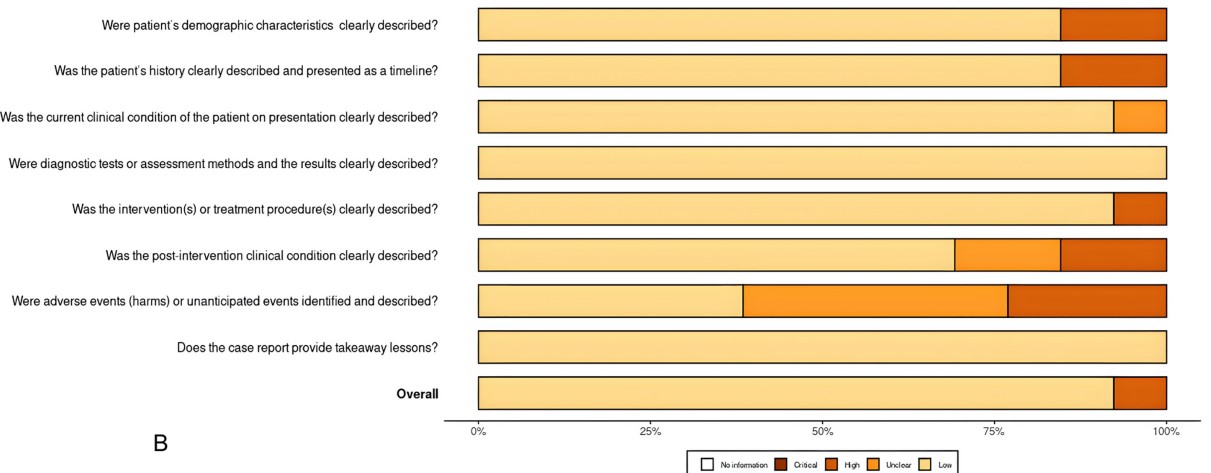

**Figure 3** **(A and B) Risk of bias summary plot for case reports.**

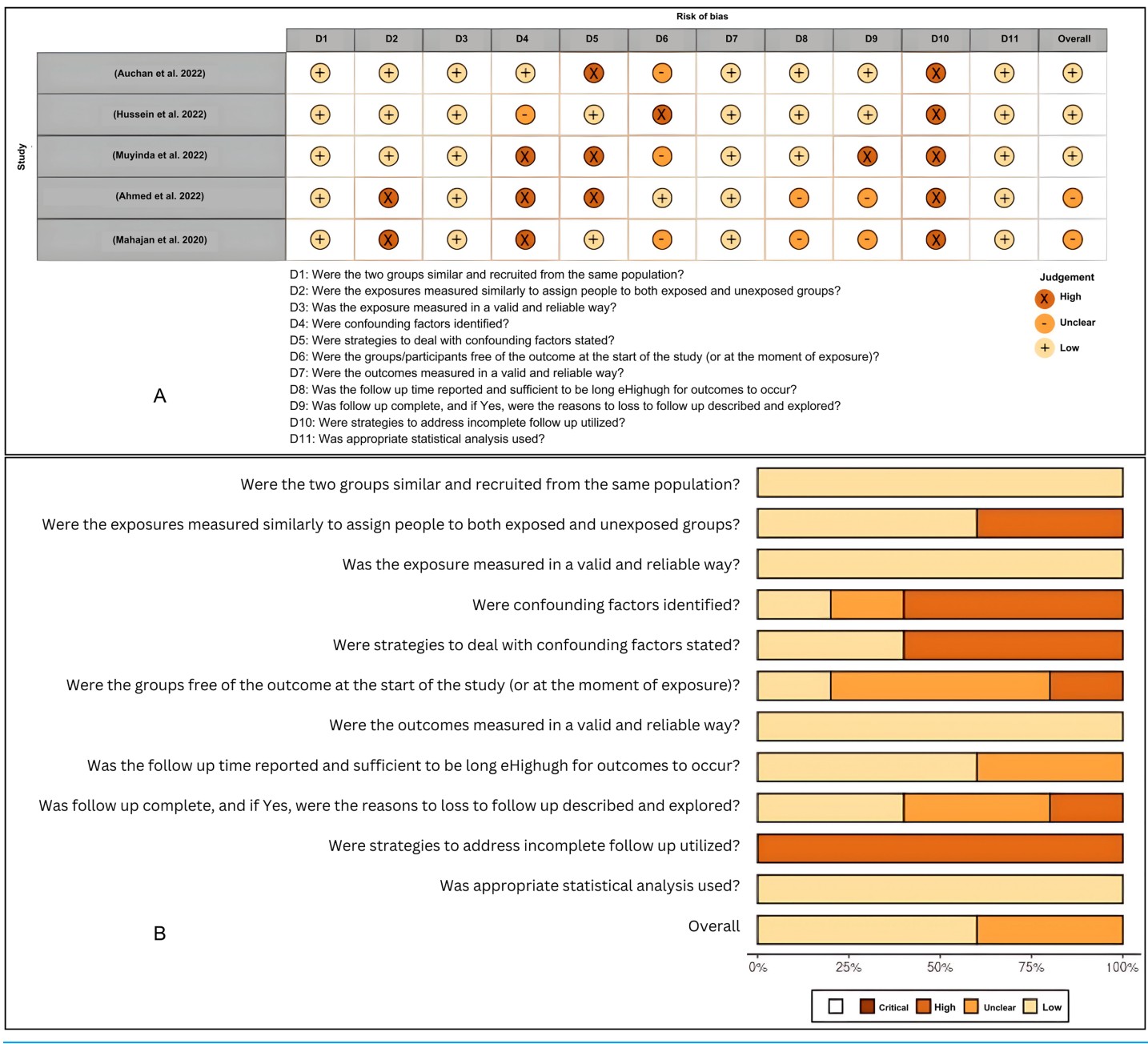

**Figure 4 (A and B) Risk of bias summary plot for cohort studies.**

(*Boonyarangka et al., 2022*; *Asmarawati et al., 2022*; *Suarez et al., 2022*). Estimated crude mortality was 10.71 and 5.87 per 1,000 person-days for patients with and without concurrent malaria, respectively (*Lee, Hong & Kim, 2022*). In a case report from India, a 28-year-old patient with co-infection died after developing severe hypoxia and cerebral malaria (*Caglar et al., 2021*).

The commonest co-morbidities recorded within the studies were diabetes Mellitus (DM) (*Boonyarangka et al., 2022*; *Lee, Hong & Kim, 2022*; *Asmarawati et al., 2022*; *Suarez et al., 2022*; *Scalisi et al., 2022*; *Chen et al., 2021*), hypertension (HTN)

**Table 4 GRADE assessment of quality and certainty of evidence.**

| No. of studies | Design | Risk of bias | Inconsistency | Indirectness[1] | Imprecision | Other[2] | Certainty (overall score)[3] |
|---|---|---|---|---|---|---|---|
| **Outcome: Length of hospital stay more than 3 days** | | | | | | | |
| 7 | Very Low (Case reports) | No serious risk of bias | No serious inconsistency | No serious | No serious | None | ⊕⊕⊕◯ Moderate |
| **Outcome: Severe disease & ICU admission** | | | | | | | |
| 10 | Low (Observational) | No serious risk of bias | No serious inconsistency | No serious | No serious | None | ⊕⊕⊕⊕ High |
| **Outcome: Mortality rate** | | | | | | | |
| 5 | Low (Observational) | No serious risk of bias | No serious inconsistency | No serious | No serious | None | ⊕⊕⊕⊕ High |
| **Outcome: Length of hospital stay fast recovery** | | | | | | | |
| 1 | Low cohort study | Likely inadequately control confounding & Follow up | No serious inconsistency | No serious | No serious | None | ⊕⊕◯◯ Low |

**Notes:**
[1] Indirectness includes consideration of
- Indirect (between study) comparisons
- Indirect (surrogate) outcomes
- Applicability (study populations, interventions or comparisons that are different than those of interest)

[2] Other considerations for downgrading include publication bias. Other considerations for upgrading include a strong association with no plausible confounders, a dose response relationship, and if all plausible confounders or biases would decrease the size of the effect (if there is evidence of an effect), or increase it if there is evidence of no harmful effect (safety)

[3] 4 ⊕⊕⊕⊕ **High** = This research provides a very good indication of the likely effect.
3 ⊕⊕⊕◯ **Moderate** = This research provides a good indication of the likely effect.
2 ⊕⊕◯◯ **Low** = This research provides some indication of the likely effect.
1 ⊕◯◯◯ **Very low** = This research does not provide a reliable indication of the likely effect.

(*Boonyarangka et al., 2022*; *Forero-Peña et al., 2022*; *Lee, Hong & Kim, 2022*; *Asmarawati et al., 2022*; *Suarez et al., 2022*; *Chen et al., 2021*), cardiovascular disease (CVD), (*Boonyarangka et al., 2022*; *Lee, Hong & Kim, 2022*; *Chen et al., 2021*), respiratory disorders that include chronic obstructive pulmonary disease (COPD), asthma, and tuberculosis (*Boonyarangka et al., 2022*; *Suarez et al., 2022*; *Chen et al., 2021*). Obesity and dyslipidemia were reported in two studies (*Boonyarangka et al., 2022*; *Forero-Peña et al., 2022*), immune deficiency, including human immunodeficiency virus (HIV) (*Boonyarangka et al., 2022*; *Suarez et al., 2022*), and malignancy was reported in two other studies (*Lee, Hong & Kim, 2022*; *Suarez et al., 2022*). One study described the co-infection of COVID-19 and Malaria in a patient with risperidone-treated autism (*Muyinda et al., 2022*). The co-morbidities, duration of hospitalization, and clinical outcomes are illustrated in Table 3.

## Quality of the included studies and publication bias

The quality of the studies and the risk of bias in individual studies were assessed according to the JBI Critical Appraisal Tools for the case report studies, cohort studies, and case series (File S2). The risk of bias for individual case reports and cohort studies as well as across all included studies, were summarized in visualization charts (Figs. 3A, 3B, 4A and 4B). Five studies (*Campbell et al., 2020*; *López-Farfán et al., 2022*; *Igala et al., 2022*; *EBioMedicine, 2021*; *Wilairatana et al., 2021*) were high-quality, whereas six studies (*JBI, 2020*; *Briggs et al., 2022*; *Kalungi et al., 2021*; *Zawawi et al., 2020*; *Konozy et al., 2022*; *Guha et al., 2021*)

were graded low to moderate quality. Due to the study design, sample size, and consistency of the data as well as the standardized tool to measure the outcomes, the level of certainty was graded low to moderate for the clinical outcomes that tested, *i.e.*, length of hospital stays, clinical severity and ICU admission and as well as mortality rate (Table 4). The review comprises 13 studies, which were case reports and one case series; this renders them of low certainty of evidence and high selection bias. The inability to address the confounding factors or strategies to adjust them was a common weakness across the cohort studies. All studies included in the review analyzed the clinical outcomes of COVID-19 and Malaria, although with different methods to measure the outcomes, clearly, and they reported the results.

## DISCUSSION

As reported in our study, most COVID-19 and malaria co-infection cases occurred in sub-Saharan African countries, specifically Sudan and Uganda, as demonstrated in significant cohort studies (*Boonyarangka et al., 2022*; *Lee, Hong & Kim, 2022*; *Asmarawati et al., 2022*; *Suarez et al., 2022*). However, most of the case report studies included in this review were from Southeast Asia, as shown in Fig. 2. However, the included case reports neither fairly reflect the malaria burden according to specific age groups or gender of the population nor demonstrate the risk factors and co-morbidities in most of them. While having a lower overall COVID-19 incidence and mortality rate than other continents, sub-Saharan Africa suffers disproportionately more from other infectious diseases such as malaria and tuberculosis (*World Health Organization, 2023*; *Kishore et al., 2020*). Therefore, a significant public health concern in Africa is the potential impact of COVID-19 on managing these diseases and the possible consequences of any clinical interactions between COVID-19 and these diseases, particularly where geographic overlap leads to high levels of co-infection. In 2020, 241 million malaria cases were estimated, along with 627,000 deaths from malaria, according to the WHO Global Malaria Report. Compared to figures for 2019, an increase of 14 million and 69,000 were recorded. Sub-Saharan Africa continues to take on the tremendous burden of the disease's impact. In addition, children under five comprise over 80% of the region's fatalities (*Mahajan et al., 2020*).

Severe malaria arises when infections are complicated by severe organ failures or abnormalities in the patient's blood or metabolism. According to our review of case reports and series in the included studies, co-infected people frequently exhibit hyper-coagulopathy, thrombocytopenia, anemia, hyper-parasitemia, renal impairment, and increased liver enzymes, indicating the possible diagnosis of severe malaria (*Huang et al., 2022*; *Boonyarangka et al., 2022*; *Achan et al., 2022*; *Muyinda et al., 2022*; *Suarez et al., 2022*; *Shahid et al., 2021*; *Jochum et al., 2021*; *Caglar et al., 2021*). The manifestations of severe malaria include the following: altered consciousness or coma, low hemoglobin, acute kidney injury, acute respiratory distress syndrome, circulatory collapse/shock, acidosis, jaundice (combined with other symptoms of severe malaria), disseminated intravascular coagulation, and hyper-parasitemia (*Global Tuberculosis Programme, 2020*). Nevertheless, most patients displaying these symptoms recovered and were discharged

from the hospital without any manifestation of severe COVID-19 infection (*Huang et al., 2022*; *Achan et al., 2022*; *Muyinda et al., 2022*; *Suarez et al., 2022*; *Shahid et al., 2021*; *Jochum et al., 2021*). The low severity of COVID-19 in such patients can be attributed to the fact that COVID-19 and malaria co-infection may enhance recovery from COVID-19, and the virus was cleared by the glycosyl-phosphatidyl-inositol antibodies against plasmodium-specific antigens which may cross-react with SARS-CoV-2 antibodies (*World Health Organization, 2021*; *Osei et al., 2022*). The antibodies against glycosyl-phosphatidyl-inositol (GPI), which are the anchor molecules of membrane proteins of Plasmodium species, were found to be associated with asymptomatic malaria (*De Mendonça & Barral-Netto, 2015*). The anti-GPI antibodies may neutralize the toxic pro-inflammatory effect triggered by Plasmodium infection. A study from Uganda concluded that individuals with low previous *P. falciparum* exposure were found to have higher rates of severe COVID-19 cases than those with high *P. falciparum* exposure (*Achan et al., 2022*). By contrast, a study conducted in India found that anti-malarial antibody levels may not be associated with the outcomes of COVID-19 (*Sethi et al., 2023*). Moreover, genetic factors are among the most effective elements in susceptibility/ resistance, outcomes, and disease progression. Consequently, the genetic protection against an infectious disease of an individual may determine the susceptibility to a life-threatening disease, and evolutionary genetic linkages through Angiotensin-Converting Enzyme 2 (ACE2) polymorphisms have been reported as an explanation for the lower burden of COVID-19 in malaria-endemic regions (*Napoli & Nioi, 2020*). The earlier analyses defined prolonged hospitalization as any hospitalization with a stay of at least 21 days (*Anderson et al., 2015*). In the primary clinical outcome of this study, almost all the patients with COVID-19 and malaria co-infection showed short-stay hospitalization of fewer than 21 days with a mean range between 11 to 17 days (*Huang et al., 2022*; *Boonyarangka et al., 2022*). The cohort study from Uganda demonstrated that patients with a positive malaria test had lower hospitalization rates than patients with a negative malaria test (*Asmarawati et al., 2022*). On the other hand, a further study from Uganda reported that 86% of co-infection patients were discharged in good condition compared to only 6% admitted to the ICU (*Boonyarangka et al., 2022*). These results agreed with the result that reported individuals who tested positive for malaria remained in the hospital for shorter periods than those who tested negative (*Ingabire et al., 2022*). Given that COVID-19 and malaria share many signs and symptoms, it is likely that some COVID-19-malaria co-infected patients had malaria as their primary illness and responded to therapy, which is typical of malaria episodes. The other possibility is that those patients might die rapidly, which could be recorded as a shorter stay at the hospital (*Ingabire et al., 2022*).

Many investigations on COVID-19 infections and mortalities have been undertaken in light of the COVID-19 pandemic. The severity of COVID-19 is thought to be exacerbated by the presence of many underlying disorders or co-infections, suggesting that underlying diseases are a crucial factor in the current COVID-19 mortalities (*Lee et al., 2022*). Our findings suggest that individuals who undergo COVID-19 and malaria co-infection have a higher risk of mortalities, as shown in the study done in Sudan, where crude mortality rates

were 10.71 and 5.87 per 1,000 people for patients with and without concurrent malaria, respectively (*Lee, Hong & Kim, 2022*). Most malaria patients in this study exhibited renal impairment, acidosis, prostration, and hyper-parasitemia during the clinical presentation; over one-third of patients came in shock, and 14.6% had pulmonary edema with hypertension and diabetes as co-morbidities (*Lee, Hong & Kim, 2022*). Also, the other Sudanese study demonstrated that the overall mortality was 40.4% in the included participants; the risk factors were associated with being male and aged between 60 and 70 years. In this study, co-infected patients' most frequent symptoms and consequences were shortness of breath and acute respiratory distress syndrome (*Suarez et al., 2022*). However, only one case report from the 13 detailed case reports indicated that the 28-year-old Indian patient co-infected with COVID-19 and malaria developed severe hypoxia and cerebral malaria and died on day four after hospitalization (*Caglar et al., 2021*). From all the initial analyses, we can suppose that most of the mortalities in COVID-19 and malaria co-infection were continually associated with co-morbidities and symptoms of severe malaria regardless of the age or sex of the patient. The finding partially agreed with a study that conveyed the clinical features and outcomes of hospitalized COVID-19 patients in one African country and concluded that younger age and non-obesity were associated with clinical improvement compared to older age. In contrast, coexisting co-morbidities were associated with mortality (*Nachega et al., 2020*). Another cohort study of children and adolescents hospitalized with COVID-19 in sub-Saharan Africa showed high morbidity and mortality rates among infants and patients with chronic co-morbidities (*Nachega et al., 2022*).

More evidence is needed on how malaria susceptibility and immune response are affected by co-infection with COVID-19 and *vice versa*. In the Uganda cohort study, prior malaria exposure decreases the percentage of severity and mortality in COVID-19 and malaria co-infection patients. In addition, there is a significant burden of *P. Falciparum* infection among patients with COVID-19 who are older than 60 years of age, even though the burden of malaria disease in malaria-endemic settings is primarily concentrated in infants and young children (*Boonyarangka et al., 2022*). It is well-comprehended that the severity of COVID-19 is linked to a rise in cytokines and chemokines (*Kunnumakkara et al., 2021*). However, this study reported that patients with *P. Falciparum* infection had greater tumor necrosis factor α (TNF-α) levels. In contrast, those with low exposure to malaria had higher interleukin 7 (IL-7) and transforming growth factor β1 (TGF-β1) levels. Those with more severe symptoms and worse outcomes had elevated IL-6, IL-10, TNF-α, and TGF-β1 levels (*Boonyarangka et al., 2022*).

The limitations of this study originate from the proportion of case report-based research included *versus* other types of observational studies. The case reports usually introduce potential selection bias because reported cases are typically individual in presentation and management. Furthermore, due to the small number of relevant research conducted, the number of included studies is limited, with a minimal number of countries covered, which makes it challenging to conclude results at this time, particularly given the clinical outcomes of malaria and COVID-19 co-infection. Although this systematic analysis offers crucial information about the clinical and public health outcomes of COVID-19 and

malaria co-infection, it is evident that additional research is required to show how immunological background, related risk factors, and therapeutic data influence these clinical outcomes.

Despite these limitations, this review underscores the criticality of early detection and treatment for individuals with co-infection of COVID-19 and malaria. Clinicians must remain vigilant in considering the possibility of co-infection, conduct appropriate diagnostic tests, and promptly initiate antiviral and anti-malarial therapies. Furthermore, patients with underlying health conditions face a heightened risk, necessitating comprehensive assessment and care. Collaboration among specialists plays a pivotal role in ensuring holistic patient management. Integration of surveillance systems enables effective detection and monitoring of co-infections, facilitating targeted interventions. It is imperative to establish robust data collection mechanisms to monitor the prevalence and outcomes of co-infection accurately. Strengthening healthcare infrastructure, including the availability of diagnostic tests and medications, is paramount. Health education campaigns should be designed to raise awareness and promote preventive measures. Further research is warranted to deepen our understanding of the interplay between co-infection, develop more effective prevention strategies, including vaccines and antiviral therapies, and explore the specific impact on vulnerable populations.

# CONCLUSIONS

It is well-reported that COVID-19 has devastatingly impacted other concomitant disorders, including infectious diseases like malaria, especially in countries where malaria is prevalent. Estimates of the clinical impact and outcomes are based on various data on the mortality and morbidity of COVID-19 and malaria co-infections. According to our data, most patients with co-infections are hospitalized for a short-term duration and have mild to moderate disease severity. However, co-morbidities coexistence is significantly associated with the high mortality in COVID-19 and malaria co-infection; furthermore, the symptoms and signs of severe malaria at presentation carry worse outcomes.

## Funding

The Deputyship for Research & Innovation, Ministry of Education in Saudi Arabia, funded this APC through the project number ISP-2024. The funders had no role in study design, data collection and analysis, decision to publish, or preparation of the manuscript.

## Grant Disclosures

The following grant information was disclosed by the authors:
Deputyship for Research & Innovation, Ministry of Education in Saudi Arabia: ISP-2024.

## Competing Interests

The authors declare that they have no competing interests.

## Author Contributions

- Amal H. Mohamed conceived and designed the experiments, authored or reviewed drafts of the article, and approved the final draft.
- Ebtihal Eltyeb conceived and designed the experiments, prepared figures and/or tables, authored or reviewed drafts of the article, and approved the final draft.
- Badria Said performed the experiments, prepared figures and/or tables, authored or reviewed drafts of the article, and approved the final draft.
- Raga Eltayeb performed the experiments, prepared figures and/or tables, and approved the final draft.
- Abdullah Algaissi analyzed the data, authored or reviewed drafts of the article, supervised the project, and approved the final draft.
- Didier Hober analyzed the data, authored or reviewed drafts of the article, supervised the project, and approved the final draft.
- Abdulaziz H. Alhazmi analyzed the data, authored or reviewed drafts of the article, supervised the project, and approved the final draft.

## Data Availability

This is a systematic review.

## Supplemental Information

Supplemental information for this article can be found online at http://dx.doi.org/10.7717/peerj.17160#supplemental-information.

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
