# Peer review of "COVID-19 and malaria co-infection: a systematic review of clinical outcomes in endemic areas"

_PeerJ, doi:10.7717/peerj.17160_

## Round 0.1 · original submission · Major Revisions

Dear Dr. Moahmmed and colleagues:

Thanks for submitting your manuscript to PeerJ. I have now received three independent reviews of your manuscript, and as you will see, the reviewers raised some concerns about the research. Despite this, these reviewers are very optimistic about your work and the potential impact it will have on research studying COVID-19 and malaria co-infections. Thus, I encourage you to revise your manuscript, accordingly, considering all the concerns raised by the reviewers.

There are many concerns pointed out by the reviewers, and you will need to address all of these and expect a thorough review of your revised manuscript by these same reviewers.

Please be sure to clarify your methods and statistics and make sure the workflow is repeatable.

I look forward to seeing your revision, and thanks again for submitting your work to PeerJ.

Good luck with your revision,

Best,

-joe

Reviewer 1 ·

Basic reporting

Raw data are not completely shared.

Experimental design

1. Search strategy: search terms are used as keywords or in the title or in the abstract? And please present the full search strategies for different databases including any filters and limits used in the supplementary files.
2. Figure 1: Ineligible records by automation tools(n=341). What are the criteria for automation exclusion? And the amount of each exclusion criterium should be added in step 2 of the PRISMA flow diagram.

Validity of the findings

1. It's suitable to apply GRADE to analyze the evidence quality evaluation. The evidence profile and the summary of findings table are needed to be presented to depict the quality of evidence as assessed by the GRADE tool.
2. Figures 3 and 4 are not completed. Please provide the correct figures for the Risk of Bias Graph and Summary. And also present the table of Risk of bias scores of included studies.
3. Table 4 misses the item of Was statistical analysis appropriate?
4. The checklist of JBI CRITICAL APPRAISAL CHECKLIST FOR COHORT STUDIES should also be presented.
5. PRISMA 2020 Checklist should show the exact Line number for each item. And some items are actually not in the manuscript such as Item 7.

Additional comments

There are some spelling and grammatical mistakes. For example, SWIM should be SWiM.

Reviewer 2 ·

Basic reporting

More recent references to give context for conducting this review are needed in the introduction. An example is in lines 58, 59, where the authors mention the epidemiological contradiction between Covid and malaria. Another example is in line 64; the treatment of Covid 19 with anti-malarial drugs was under controversy and studies were not conclusive.

The contributions that makes this new review in light of the previously published reviews in this topic are not clearly mentioned in the introduction. I suggest improve all this to provide more justification for the study.

Figure 3 resolution is bad

Experimental design

This is a systematic review to address the clinical outcomes of Covid-19 and malaria co-infection that includes cohort studies, case reports and case series. Despite this question is of utmost importance for global health policies there are not so many comprehensive studies to drawn conclusions. This review contributes to synthesize the existing studies. However, the epidemiological implications are not well stated, the prevalence of malaria among Covid patients or vice versa was not calculated and statistics could be applied to calculate the frequency of co-morbidities and outcomes of the co-infection.

Validity of the findings

The authors you should expand upon the knowledge gap being filled.

Statistical analysis (as I have noted above) should be included

The conclusions are appropriately stated, supported for the results and acknowledge the limitations of the data.

Additional comments

In the abstract is stated: “Out of 1596 screened articles, 19 met the inclusion
Criteria” (line 23). However, actually there are not 1596 original articles because 1064 were duplicates (line 131). This needs to be clarified to avoid to confuse the readers. It could be said: 1596 records were identified in the scientific databases.

·

Basic reporting

NA

Experimental design

NA

Validity of the findings

NA

Additional comments

The manuscript entitled “COVID-19 and Malaria Co-infection: A Systematic Review of Clinical Outcomes and Epidemiological Implications in Endemic Areas” addresses a highly relevant and timely topic, especially considering the current global health scenario. The systematic approach to data collection, utilizing multiple databases, is commendable. This comprehensive data collection and an in-depth discussion that provides context by comparing the findings with existing literature adds significant value to the manuscript. However, I believe the manuscript needs some correction/improvement before it gets accepted in the journal.

1) Typo/grammatical error
The authors should go through the manuscript and correct a few typos and grammatical errors; here are some examples:
a) A structured format based on Preferred Reporting Items for Systematic Review and Meta-Analyses (PRISMA) Guidelines and a checklist was were used to select and review studies included in the review. (see line no. 84)
b) including the number of patients who were diagnosed with both Malaria and COVID19 COVID-19. (Please see line no. 112)
c) One study reported 6% of ICU admissions (please see line no. 167).
d) Malaria con-infection co-infection (please see line no. 95)
e) Rewrite the sentence “ The results agreed with the results that …..” (Please see line 238)

2) The authors should add y-axis label to figure 2.
3) Text labels on both x and y axis are not clear in Figures 3 and 4.
4) The Author’s reference in table 1 is not uniform format; please use the standard format. (Please see row 3, column 1, table 1, Zhu, Min, et al. should be Zhu et al.)
5) The authors should elaborate and explain the rationale for selecting 35 papers for comprehensive evaluation, 19 for systemic review, and so on.
6) Why non-English literature/studies were excluded from the study? The authors should also explain the reason, whether because of the language limitation or some demographic or regional reason.
7) If authors could include a figure (like a pie chart) to describe the demographic aspects mentioned in lines 141-145.
8) The authors should also clearly mention the technique RT-PCR used, not qPCR, as both approaches were used to detect the virus. If it’s the qPCR, do the studies mentioned have some information about the viral load of SARAS-Cov2?
9) In the discussion section, the authors mentioned that “ The low severity of COVID-19 in such patients can be attributed to the fact that COVID-19 and malaria co-infection may enhance recovery from COVID-19, and the virus was cleared by the glycosyl-phosphatidyl-inositol antibodies against plasmodium-specific antigens which may cross-react with SARS-CoV-2 antibodies [52].
The reference cited here is a study done on HCW only. Authors should also mention other studies conducted, if any, on the general population.
10) Also, in light of other studies, authors should discuss the statement that glycosyl-phosphatidyl-inositol antibodies against plasmodium-specific antigens can effectively clear the COVID-19 virus.
11) In lines 248-250, Did the authors also perform individual or cohort case studies, if yes, kindly also cite it here. Otherwise, authors should rewrite these lines, as the term “finding” is a bit ambiguous in this context.
12) In lines 289-290, Despite these limitations, this review underscores the criticality of early detection and treatment for individuals with co-infection of COVID-19 and Malaria.
The authors should also elaborate and explain further.

---

## Round 0.2 · Minor Revisions

Dear Dr. Moahmmed and colleagues:

Thanks for revising your manuscript. The two reviewers willing to re-review are mostly satisfied with your revision (as am I). Great! However, there are some additional issues to entertain. Please address these ASAP so we may move forward with your manuscript.

Importantly, please provide all relevant information in your figures, making sure they are stand-alone (with their cognate legends). Please ensure that your workflow is repeatable. Please add missing information pointed out by the reviewers.

Good luck with your revision,

Best,

-joe

Reviewer 1 ·

Basic reporting

no comments

Experimental design

Figure 1: Ineligible records by automation tools (n=341). What are the criteria for automation exclusion? And the amount of each exclusion criterion should be added in step 2 of the PRISMA flow diagram.
The authors responded that criteria for automation exclusion was addressed in PRISMA flow diagram.
Sorry, I didn't find this in the manuscript.

Validity of the findings

1. Figures 3 and 4. The risk of bias summary plot shows the risk for each of the included studies, while the risk of graph bias shows the risk of bias items presented as percentages across all included studies. Refer Zhang NN, Qu FJ, Liu H, Li ZJ, Zhang YC, Han X, Zhu ZY, Lv Y. Prognostic impact of tertiary lymphoid structures in breast cancer prognosis: a systematic review and meta-analysis. Cancer Cell Int. 2021 Oct 15;21(1):536. doi: 10.1186/s12935-021-02242-x. PMID: 34654433; PMCID: PMC8520238.
2. Table 4 misses the item of Was statistical analysis appropriate? In the revised manuscript, Table 4 was moved to supplementary. But this item was not added.
3. Sorry, I did't see the checklist of JBI Critical Appraisal Checklist for COHORT STUDIES.
4. PRISMA 2020 Checklist still didn't show the exact Line number for each item. For example, Item 7 Search strategy is in Lines 96-105, but it is 106-125 in the checklist.

Additional comments

no comments.

·

Basic reporting

NA

Experimental design

NA

Validity of the findings

NA

Additional comments

The revised manuscript on the co-infection of COVID-19 and Malaria in endemic areas has been thoroughly enhanced, with the authors diligently addressing nearly all comments and suggestions. Given the comprehensive analysis presented, I recommend the manuscript titled "COVID-19 and Malaria Co-infection: A Systematic Review of Clinical Outcomes in Endemic Areas" for publication.

---

## Round 0.3 · accepted · Accept

Dear Dr. Moahmmed and colleagues:

Thanks for revising your manuscript based on the concerns raised by the reviewers. I now believe that your manuscript is suitable for publication. Congratulations! I look forward to seeing this work in print, and I anticipate it being an important resource for groups studying COVID-19 and malaria co-infections. Thanks again for choosing PeerJ to publish such important work.

Best,

-joe